# Soft Gates for Sharp Experts in Tabular Representation Learning

## Abstract

Neural networks consistently underperform gradient-boosted trees on tabular data, yet the structural reasons remain poorly understood. We design the Sparse Feature Routing Network (SFR Net)—not as a benchmark entry, but as an *experimental apparatus*—to test three hypotheses about tabular inductive biases: **(H1)** per-feature experts improve over shared encoders even with fewer parameters, with gains amplified by instance-wise routing; **(H2)** instance-wise sparsity helps only when differentiable—hard gating collapses optimization; **(H3)** the learned routing produces faithful attributions confirmed by deletion tests against random baselines. The most striking finding: hard sparsity degrades accuracy *below* the dense baseline, while entropy-regularized softmax achieves extreme sparsity (2.9 of 14 effective features) *and* highest accuracy—soft gates produce sharp experts; hard gates produce dead ones. Controlled ablations and generalization across 13 benchmarks and 12 baselines yield testable design principles for tabular architectures.

## 1 Introduction

Deep models often lose to gradient-boosted decision trees (GBDTs) on tabular benchmarks (Grinsztajn et al., 2022; Gorishniy et al., 2021; 2025), yet most work asks *what* architecture to use rather than *why* certain inductive biases succeed. We take a science-first approach: tabular data—with semantically heterogeneous columns—offers a testbed where architectural assumptions can be cleanly isolated. Unlike image patches or tokens that share a generative process, "age," "zip code," and "blood pressure" have distinct distributions, scales, and roles. We hypothesize this heterogeneity is a *structural property* that architectures should reflect, and formulate three testable claims: **(H1)** per-feature expert networks outperform shared encoders, with gains amplified by instance-wise routing; **(H2)** instance-wise sparsity helps only when smooth—hard gating (top-$k$, entmax) disrupts gradient flow and kills expert specialization; **(H3)** the sparse router's selections are faithful, confirmed by deletion experiments against random baselines.

To test these we introduce **SFR Net**: per-feature expert MLPs, an entropy-regularized softmax gate, and a low-rank interaction head—designed for clean ablations (disable routing for H1, swap gating for H2, delete top features for H3). The tension between soft and hard expert selection is well-studied in mixture-of-experts (MoE) models for language and vision (Shazeer et al., 2017; Fedus et al., 2022), where load-balancing losses mitigate expert collapse. Our setting differs: experts are *per-feature*, not per-token, so routing directly reflects which input dimensions matter for each instance. Two tabular-specific architectures are directly relevant: TabNet (Arik & Pfister, 2021) uses sparsemax (hard sparsity) for instance-wise selection; NAMs (Agarwal et al., 2021) enforce global additive per-feature structure without instance-wise selection. Our ablations test whether soft routing and instance-wise selection matter.

## 2 Experimental Apparatus: SFR Net

Each hypothesis maps to a component that can be independently ablated (diagram in Appendix A).

**Feature-wise experts (H1).** Each feature $x_j$ is processed by a dedicated small MLP $E_j \to \mathbf{h}_j \in \mathbb{R}^D$, with type-specific architectures for numerical and categorical features.

**Entropy-regularized router (H2).** A scoring network produces importance $s_j$ per expert output, normalized via softmax into $\boldsymbol{\alpha}$. Sparsity is induced by minimizing entropy:

$$\mathcal{L} = \mathcal{L}_{\text{task}} + \lambda\, H(\boldsymbol{\alpha}), \quad \text{where } H(\boldsymbol{\alpha}) = -\sum_j \alpha_j \log \alpha_j. \tag{1}$$

This is fully differentiable—all experts receive gradients—while producing peaked distributions. We measure sparsity via *effective active features* $F_{\text{eff}} = \exp(H(\boldsymbol{\alpha}))$: uniform $\to F$; delta $\to 1$.

**Low-rank interaction head.** Weights $\boldsymbol{\alpha}$ gate an additive path $\mathbf{r}^{(1)} = \sum_j \alpha_j \mathbf{h}_j$ and a second-order path $\mathbf{r}^{(2)} = \sum_j \alpha_j (\mathbf{k}_j \odot \mathbf{v}_j)$; their concatenation feeds a prediction head.

## 3 HYPOTHESES AND EVIDENCE

All experiments use standardized train/val/test splits with 5 seeds; we report means throughout (full ±std in Appendix C). On our primary ablation benchmark, per-seed standard deviations are below 0.2pp for accuracy and 0.005 for AUC, so all H1–H3 gaps ≥0.5pp reflect robust differences; the H2 soft-vs.-hard contrast (2–4pp) exceeds variance by an order of magnitude. The three hypotheses form a progression: H1 identifies *what* helps (decomposition + routing), H2 identifies *how* routing must be implemented (soft, not hard), and H3 validates that the resulting selections are trustworthy. We report detailed ablations on a binary classification benchmark ($F = 14$ features: 6 continuous, 8 categorical) and verify generalization across 13 datasets spanning classification and regression. Hyperparameters in Appendix D; cross-dataset results in Appendices B–C.

### 3.1 H1: SPECIALIZATION HELPS—ROUTING AMPLIFIES IT

Table 1: **H1**. Per-feature decomposition improves over a shared MLP that has $5.4\times$ more parameters, ruling out a capacity explanation. Routing and low-rank interaction provide a further joint gain. Cross-dataset results in Appendix B.

| Configuration | Acc (%) | AUC | Params |
|---|---|---|---|
| Standard MLP (shared) | 85.40 | .9085 | 94K |
| Decomposed MLP (per-feature; $\equiv$ NAM) | 85.86 | .9128 | <17K |
| SFR Net (experts + routing + interaction) | **86.19** | **.9153** | 17K |

The Decomposed MLP (Table 1, row 2) averages expert outputs and feeds them to a prediction head—no router, no interaction module. This is structurally equivalent to a NAM (Agarwal et al., 2021) with nonlinear experts, so it serves as a direct NAM baseline. Per-feature decomposition alone yields +0.46pp over the standard MLP despite using $5.4\times$ fewer parameters (<17K vs. 94K), ruling out both a capacity explanation *and* the possibility that the 94K MLP simply overfits. The contribution of H1 is not the claim that per-feature processing helps—this is established (Agarwal et al., 2021; Arik & Pfister, 2021)—but the capacity-controlled evidence that the gain is inductive, not parametric. SFR Net adds sparse routing and low-rank interaction (+0.33pp jointly); routing *enables* interaction by concentrating it on the selected features, making the two components complementary rather than independently separable. We note that this modest incremental gain is *not* the central finding about routing—that is H2, where the choice between soft and hard gating mechanisms produces a 4pp swing, the paper's largest effect.

## 3.2 H2: SPARSITY HELPS, BUT ONLY WHEN DIFFERENTIABLE

Table 2: **H2**: Routing ablation. Hard gating degrades below the dense baseline; soft entropy routing achieves best accuracy *and* highest sparsity ($F_{\text{eff}}$=2.90 of 14).

| Routing | Acc (%) | AUC | $F_{\text{eff}}$ |
|---|---|---|---|
| No routing (Avg Pool) | 85.86 | .9128 | 14 (all) |
| Top-$k$=5 (hard) | 81.67 | .8477 | 5.0 |
| Entmax ($\alpha$=1.5) | 83.34 | .8811 | $\sim$5 |
| Softmax ($\tau$=2, no reg.) | 86.17 | .9152 | $\sim$8 |
| **Softmax + entropy** | **86.19** | **.9153** | **2.90** |

Hard gating drops accuracy 2–4pp *below* the dense baseline—a catastrophic failure, not merely a suboptimal configuration. The gap far exceeds per-seed variance by an order of magnitude. Table 2 constitutes a dose–response curve along the *soft→hard* axis: performance degrades monotonically with gating discreteness (softmax+entropy > softmax > entmax ≫ top-$k$), and the transition from soft to hard crosses the dense baseline—evidence that the failure is the discrete mechanism, not the sparsity level.

The mechanism is analytically verifiable. Under top-$k$, $\alpha_j = 0$ for $j \notin$ top-$k$, so $\partial\mathcal{L}/\partial\mathbf{h}_j = \alpha_j \cdot \partial\mathcal{L}/\partial\mathbf{r} = \mathbf{0}$: excluded experts receive *exactly zero* gradient regardless of their output quality. Entmax ($\alpha > 1$) produces exactly sparse outputs, creating the same zero-gradient condition. Under softmax, $\alpha_j > 0$ for all $j$, guaranteeing $\partial\mathcal{L}/\partial\mathbf{h}_j \neq \mathbf{0}$ universally. That hard gates zero gradients is elementary; what is *not* obvious a priori is that this elementary property produces catastrophic failure (2–4pp below dense) in the tabular setting, while the soft alternative achieves extreme sparsity ($F_{\text{eff}}$=2.90) *and* best accuracy. The critical insight: entropy regularization decouples output sparsity from gradient sparsity, concentrating the output distribution while preserving gradient flow to all experts.

Sparsity thus emerges as a *consequence* of training, not a constraint imposed on it. The regularization strength $\lambda$ controls the trade-off: $\lambda$=0 yields $F_{\text{eff}}\approx 8$ (diffuse), $\lambda$=0.01 compresses to $F_{\text{eff}}$=2.90 with no accuracy loss, and $\lambda$=0.1 over-regularizes (not shown). This reframes prior negative results on sparse tabular routing (Gorishniy et al., 2021): the failure may be the hard mechanism (as in TabNet's sparsemax (Arik & Pfister, 2021)), not sparsity itself.

## 3.3 H3: ROUTING PRODUCES FAITHFUL ATTRIBUTIONS

For each test instance we identify the top-3 features by routing weight and measure AUC after zeroing them out, compared against a **random baseline** (3 random features, 50 draws). We test two entropy regimes to examine how sparsity affects attribution quality.

Table 3: **H3**: Deletion test. Sparser routing ($\lambda$=0.01, $F_{\text{eff}}$=2.90) yields more concentrated—and more faithful—attributions. Ratio = $\Delta\text{AUC}_{\text{top}}/\Delta\text{AUC}_{\text{rand}}$ relative to baseline.

| $\lambda$ | $F_{\text{eff}}$ | Baseline AUC | AUC after deletion Random-3 | Top-3 by router | Ratio |
|---|---|---|---|---|---|
| 0 (no reg.) | 4.48 | 0.914 | 0.846 | 0.684 | 3.4$\times$ |
| 0.01 | 2.90 | 0.904 | 0.836 | **0.526** | **5.6$\times$** |

With $\lambda$=0.01, deleting the top-3 routed features collapses AUC to near-chance (0.526), a 5.6$\times$ larger drop than random deletion—confirming that routing weights identify features the model *actually relies on*. The deletion test itself is standard methodology; the finding is the *dose–response* between sparsity and faithfulness: as $\lambda$ increases and $F_{\text{eff}}$ drops from 4.48 to 2.90, the ratio jumps from 3.4$\times$ to 5.6$\times$. This co-variation suggests that entropy regularization does not merely compress representations but forces the router to make sharper, more decision-relevant selections. We note

this establishes *attribution faithfulness* (weights reflect the model's internal computation), not causal importance in the data-generating sense—a distinction shared by all gradient-based and attention-based attribution methods.

## 3.4 GENERALIZATION ACROSS 13 BENCHMARKS

Table 4: Average rank across 13 datasets and 12 models (full results in Appendix C). SFR Net ranks first on 5 datasets with $5$–$24\times$ fewer parameters than alternatives.

| Model | Avg Rank | #1st | #Top-3 | Params |
|---|---|---|---|---|
| SFR Net | **2.0** | **5** | **11** | 17K |
| TabM | 2.4 | 5 | 10 | 94K |
| GBDT | 3.5 | 2 | 8 | – |
| MNCA | 4.4 | 0 | 2 | – |
| TabR | 5.2 | 1 | 4 | – |
| FT-Transformer | 6.5 | 0 | 1 | 412K |
| MLP | 9.3 | 0 | 0 | 94K |

While the controlled ablations above test specific hypotheses, the same architectural priors yield strong generalization: SFR Net's 17K parameters achieve rank 2.0 across 13 datasets, ahead of 94K-parameter TabM and 412K-parameter FT-Transformer—evidence that matching priors to data structure is more efficient than scaling capacity. A Wilcoxon signed-rank test on per-dataset ranks confirms SFR Net outperforms GBDT ($p=0.06$, 9 wins / 4 losses); the gap over TabM is directional but not significant ($p=0.18$, 7/6), consistent with the models exploiting complementary priors.

**When does routing help less?** SFR Net's two weakest results are California Housing (rank 5, $F=8$) and Diamonds (rank 4, $F=9$)—both low-dimensional regression tasks won by retrieval-based (TabR) or ensemble (TabM) methods. With few features, there is less heterogeneity to route over, and dense pairwise interactions matter more than sparse selection. This suggests a boundary condition: feature-level routing is most beneficial when $F$ is moderate to large and features are semantically diverse.

## 3.5 DISCUSSION AND FUTURE DIRECTIONS

Our results support a structural account of tabular difficulty: feature heterogeneity rewards per-feature specialization, but the larger gains come from instance-wise routing—mirroring how trees select different split features per node. The central design principle is that *sparsity should emerge from training, not be imposed a priori*. The fact that this principle succeeds with $5$–$24\times$ fewer parameters than dense alternatives suggests that matching architectural priors to data structure is fundamentally more efficient than scaling capacity.

The consistency of these findings across our benchmark suite (Table 4) suggests the underlying principles are general, not dataset-specific. The gradient analysis in Section 3.2 demonstrates analytically that hard gates produce exactly zero gradients for excluded experts—gradient starvation is a mathematical property of the architecture, not an empirical conjecture. Two natural extensions remain: (i) testing Gumbel-Softmax and straight-through estimators to isolate hard *selection* from non-differentiability, and (ii) tracking per-expert utilization across training to visualize the temporal dynamics of the collapse that the gradient analysis predicts. The deletion test follows standard practice in the attribution literature; comparing router attributions against Shapley values or LIME on synthetic data with known ground-truth importance would establish whether routing recovers true feature relevance beyond internal faithfulness. Finally, the success of feature-level routing on moderate-dimensional benchmarks motivates scaling to settings with hundreds of features and integrating sparse routing into tabular foundation model architectures.

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

## A  ARCHITECTURE DIAGRAM

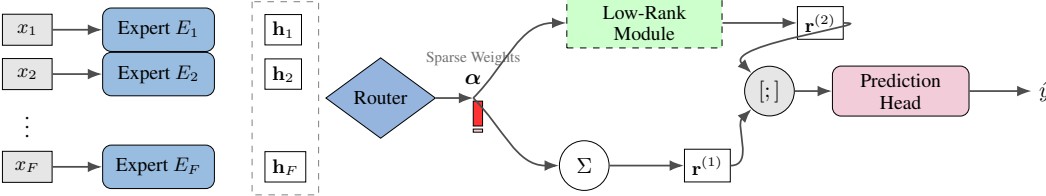

Figure 1: **SFR Net.** Feature experts (H1), entropy-regularized router (H2), routing weights as deletion targets (H3).

## B  H1: CROSS-DATASET RESULTS

Table 5 shows SFR Net compared against shared-encoder baselines across four additional datasets. The per-feature decomposition advantage transfers consistently across tasks and scales.

Table 5: SFR Net vs. shared encoders across tasks (mean ± std, 5 seeds).

| Dataset | MLP | ResNet | DCN-v2 | FT-Trans | SFR Net |
|---|---|---|---|---|---|
| **Churn** (Acc ↑) | 0.855±.003 | 0.855±.004 | 0.857±.002 | 0.859±.003 | **0.869**±.002 |
| **Otto** (Acc ↑) | 0.818±.002 | 0.817±.002 | 0.806±.002 | 0.813±.003 | **0.835**±.003 |
| **Microsoft** (RMSE ↓) | 0.748±.000 | 0.747±.000 | 0.750±.000 | 0.746±.001 | **0.735**±.001 |
| **Covtype** (Acc ↑) | 0.963±.001 | 0.964±.001 | 0.962±.002 | 0.970±.001 | **0.979**±.000 |

## C  FULL PER-DATASET RESULTS

Tables 6 and 7 report complete results across all 13 datasets and 12 baselines, from which the ranks in Table 4 are computed.

Table 6: Classification results (Accuracy ↑, mean ± std, 5 seeds). Bold = best; underline = second.

|         | MLP       | ResNet    | DCN2      | AutoInt   | Mixer     | SAINT     | FT-T      | TabR      | MNCA      | TabM      | GBDT      | SFR       |
|---------|-----------|-----------|-----------|-----------|-----------|-----------|-----------|-----------|-----------|-----------|-----------|-----------|
| Churn   | .855±.003 | .855±.004 | .857±.002 | .861±.005 | .859±.004 | .860±.003 | .859±.003 | .860±.003 | .860±.003 | .861±.003 | .861±.002 | **.869**±.002 |
| Adult   | .854±.002 | .855±.001 | .858±.001 | .859±.002 | .860±.001 | .860±.002 | .859±.002 | .865±.002 | .868±.002 | .863±.000 | **.872**±.001 | .869±.001 |
| Credit  | .774±.004 | .772±.003 | .770±.003 | .774±.005 | .775±.004 | .774±.005 | .775±.004 | .773±.004 | .774±.003 | **.776**±.004 | .771±.003 | .776±.003 |
| Higgs   | .718±.003 | .726±.002 | .716±.003 | .724±.003 | .725±.002 | .724±.002 | .728±.002 | .722±.001 | .726±.002 | **.739**±.002 | .726±.001 | .731±.002 |
| Covtype | .963±.001 | .964±.001 | .962±.002 | .961±.002 | .966±.002 | .967±.001 | .970±.001 | .974±.001 | .972±.000 | .974±.000 | .971±.000 | **.979**±.000 |
| Otto    | .818±.002 | .817±.002 | .806±.002 | .805±.003 | .809±.004 | .812±.002 | .813±.003 | .818±.002 | .828±.001 | .828±.001 | .832±.001 | **.835**±.003 |
| Jannis  | .784±.002 | .792±.002 | .771±.003 | .793±.002 | .793±.003 | .797±.003 | .794±.003 | .798±.002 | .799±.002 | **.808**±.002 | .801±.001 | .801±.003 |
| Wine    | .778±.015 | .771±.014 | .749±.015 | .775±.014 | .777±.015 | .768±.014 | .776±.013 | .794±.011 | .791±.014 | .794±.012 | .799±.013 | **.801**±.014 |
| BrstW   | .968±.005 | .971±.004 | .965±.006 | .970±.005 | .972±.004 | .971±.005 | .974±.003 | .976±.004 | .978±.003 | .979±.002 | **.993**±.002 | .981±.004 |

Table 7: Regression results (RMSE ↓, mean ± std, 5 seeds). Bold = best; underline = second.

|       | MLP       | ResNet    | DCN2      | AutoInt   | Mixer     | SAINT     | FT-T      | TabR      | MNCA      | TabM      | GBDT      | SFR       |
|-------|-----------|-----------|-----------|-----------|-----------|-----------|-----------|-----------|-----------|-----------|-----------|-----------|
| CA    | .495±.006 | .492±.003 | .497±.012 | .468±.006 | .475±.006 | .468±.005 | .464±.005 | **.403**±.002 | .424±.001 | .441±.001 | .427±.000 | .456±.004 |
| House | 3.11±.03  | 3.11±.03  | 3.33±.09  | 3.22±.04  | 3.19±.05  | 3.24±.06  | 3.18±.05  | 3.07±.04  | 3.09±.03  | **3.00**±.01 | 3.11±.00  | 3.04±.01  |
| Msft  | .748±.000 | .747±.000 | .750±.000 | .748±.001 | .748±.001 | .763±.007 | .746±.001 | .750±.001 | .746±.000 | .743±.000 | .741±.000 | **.735**±.001 |
| Diam  | .140±.001 | .140±.003 | .142±.003 | .139±.001 | .140±.003 | .137±.002 | .138±.001 | .133±.001 | .137±.002 | **.131**±.001 | .133±.000 | .135±.002 |

## D  HYPERPARAMETERS AND PROTOCOL

All models are trained with Adam, learning rate selected from $\{10^{-3}, 3 \times 10^{-4}, 10^{-4}\}$ via validation. Expert hidden dimension $D = 32$, expert depth = 2 layers with GELU activations. Entropy regularization $\lambda$ selected from $\{0, 0.001, 0.01, 0.1\}$. Low-rank interaction rank = 8. Batch size 256, early stopping with patience 20. The GBDT baseline is CatBoost, tuned via Optuna (100 trials) over learning rate, depth, L2 regularization, and bagging temperature, following the protocol of Gorishniy et al. (2021). All datasets use the splits from Gorishniy et al. (2021) where available.

