# OpenReview forum: "Soft Gates for Sharp Experts in Tabular Representation Learning"
_ICLR.cc/2026/Workshop/Sci4DL — Sci4DL 2026_

### Official Review · Reviewer_birS · 2026-02-12

**Fit:** 1
**Significance:** 1
**Confidence:** 1

**Summary:**

- This paper proposes an architecture for tabular data: feature-wise (by H1) 2-layer MLP experts, mixed by weight $\alpha$ through a certain scoring network. During training $\alpha$ is regularized by its entropy (H2) and the paper argues that regularization works better (H3).

**Strengths:**

- It shows excellent performance over the baseline models.
- It shows excellent novelty with justified motivations.
- Hypotheses that motivates the architecture are grounded with ablation experiments.

**Suggestions:**

- Reproducibility remains unclear, since the definitions of $q,k$ and the scoring network are missing and no code/psudocode is provided—this is my main concern.
- Following the previous point, I don't understand why the term with $q,k$ is named as low-rank “interaction”, but no cross term ($q_i,k_j$) appears—so again the definition of $q,k$ should be made explicit.
- I wonder on small-scale data would it underperform TabPFN; can this paper be used to improve TabPFN?

**Comments:**
- I believe this paper is a valuable contribution to tabular data-specific workshops.

---

### Official Review · Reviewer_TtEg · 2026-02-18

**Fit:** 2
**Significance:** 2
**Confidence:** 2

**Summary:**

This work studies feature-wise MoE architectures for tabular learning and proposes the Sparse Feature Routing Network (SFR Net), which combines per-feature expert networks with an entropy-regularized softmax router and a low-rank interaction module. Through controlled experiments, the authors show that differentiable soft routing can achieve highly sparse feature selection without harming optimization, whereas hard routing leads to expert collapse due to zero gradients. Results across multiple datasets demonstrate competitive performance with strong sparsity and suggest that routing weights provide meaningful feature importance signals.

**Strengths:**

The paper is easy to follow and well written. Its main strength is the clear hypothesis-driven experimental design: the authors formulate concrete hypotheses and use controlled ablations that isolate individual components (per-feature experts, routing mechanisms, interaction modules) to test them. This careful empirical methodology aligns well with the goals of the workshop by contributing to the understanding of deep learning through the scientific method rather than merely pursuing SOTA performance.

**Suggestions:**

The experimental details are currently insufficient for full reproducibility. Important architectural and training specifics—such as the exact router architecture, categorical feature handling, interaction module parameterization, and full hyperparameter configurations—are either missing or only partially described. Providing complete implementation details and releasing code would strengthen the paper.

The related work discussion, particularly around mixture-of-experts (MoE) and sparse routing literature, should be more comprehensive and careful. Many of the hypotheses explored in the paper (e.g., routing collapse, entropy effects, soft vs. hard gating tradeoffs) have been studied extensively in prior MoE research. The paper would benefit from a clearer positioning of what is genuinely new relative to existing work and a more explicit comparison to prior findings, especially to avoid overstating novelty.

The comparison between hard and soft gating should also be conducted more carefully. The observed performance gap may reflect optimization difficulties rather than a fundamental limitation of hard gating itself. In modern MoE systems, various techniques—such as load-balancing losses, noisy routing, straight-through estimators, or soft-to-hard annealing—are commonly used to mitigate gradient starvation. Including experiments with such techniques would help determine whether the conclusions about hard gating are robust or primarily driven by training choices.

Finally, the paper would benefit from deeper analysis of training dynamics to support the proposed mechanism. For example, tracking expert utilization, routing entropy, and gradient norms over the course of training could provide more direct evidence for the claimed "dead expert" phenomenon and clarify how entropy regularization stabilizes learning.

---

### Official Review · Reviewer_EyaB · 2026-02-26

**Fit:** 2
**Significance:** 3
**Confidence:** 1

**Summary:**

The paper empirically explores architectural properties in tabular learning. They show evidence for the following hypotheses: (i) A collection of per-feature experts is more parameter-efficient than models that share encodings for all features; (ii) In the expert-routing mechanism, it is better to have soft-routing with entropy-regularization, rather than hard-routing (e.g. top-k); (iii) in the soft-routing case, the learned routing can be robustly attributed to features. As a test-bed for these assertions, they design and experiment with the proposed SRR architecture.

**Strengths:**

The paper is written in a clean manner, with crisp and thoughtful experiments. All three hypotheses proposed in the paper are confirmed with ample empirical results. I appreciate the comparison between performance gaps and standard deviations between runs -- a somewhat obvious but often missed point. I also appreciate the remark on which tasks are SFR's weaknesses and why.

**Suggestions:**

Judging the overall significance of the work within the field of tabular learning is outside my domain of expertise -- however, as a standalone work, I could find no obvious weaknesses that need addressing.

---

### Meta-Review · Area_Chair_qSHV · 2026-03-01

**Recommendation:** Accept

**Metareview:**

I recommend accept.

---

### Decision · Program_Chairs · 2026-03-02

Accept